# Unified Interpretation of Smoothing Methods for Negative Sampling Loss Functions in Knowledge Graph Embedding

## Abstract

Knowledge Graphs (KGs) are fundamental resources in knowledge-intensive tasks in NLP. Due to the limitation of manually creating KGs, KG Completion (KGC) has an important role in automatically completing KGs by scoring their links with KG Embedding (KGE). To handle many entities in training, KGE relies on Negative Sampling (NS) loss that can reduce the computational cost by sampling. Since the appearance frequencies for each link are at most one in KGs, sparsity is an essential and inevitable problem. The NS loss is no exception. As a solution, the NS loss in KGE relies on smoothing methods like Self-Adversarial Negative Sampling (SANS) and subsampling. However, it is uncertain what kind of smoothing method is suitable for this purpose due to the lack of theoretical understanding. This paper provides theoretical interpretations of the smoothing methods for the NS loss in KGE and induces a new NS loss, Triplet-based SANS (T-SANS), that can cover the characteristics of the conventional smoothing methods. Experimental results on FB15k-237, WN18RR, and YAGO3-10 datasets showed the soundness of our interpretation and performance improvement by our T-SANS.

## 1 Introduction

Knowledge Graphs (KGs) represent human knowledge using various entities and their relationships as graph structures. KGs are fundamental resources for knowledge-intensive applications like dialog (Moon et al., 2019), question answering (Reese et al., 2020), named entity recognition (Liu et al., 2019), open-domain questions (Hu et al., 2022), and recommendation systems (Gao et al., 2020), etc.

However, to create complete KGs, we need to consider a large number of entities and all their possible relationships. Taking into account the explosively large number of combinations between entities, only relying on manual approaches is unrealistic to make complete KGs.

Knowledge Graph Completion (KGC) is a task to deal with this problem. KGC involves automatically completing missing links corresponding to relationships between entities in KGs. To complete the KGs, we need to score each link between entities. For this purpose, current KGC commonly relies on Knowledge Graph Embedding (KGE) (Bordes et al., 2011). KGE models predict the missing relations, named link prediction, by learning structural representations. In the current KGE, models need to complete a link (triplet) $(e_i, r_k, e_j)$ of entities $e_i$ and $e_j$, and their relationship $r_k$ by answering $e_i$ or $e_j$ from a given query $(?, r_k, e_j)$ or $(?, r_k, e_j)$, respectively. Hence, KGE needs to handle a large number of entities and their relationships during its training.

To handle a large number of entities and relationships in KGs, Negative Sampling (NS) loss (Mikolov et al., 2013) is frequently used for training KGE models. NS loss is originally proposed to approximate softmax cross-entropy loss to reduce computational costs by sampling false labels from its noise distribution in training. Trouillon et al. (2016) import the NS loss from word embedding to KGE with utilizing uniform distribution as its noise distribution. Sun et al. (2019) extend the NS loss to Self-Adversarial Negative Sampling (SANS) loss for efficient training of KGE. Unlike the NS with uniform distribution, the SANS loss utilizes the training model's prediction as the noise distribution. Since the negative samples in the SANS loss become more difficult to discriminate

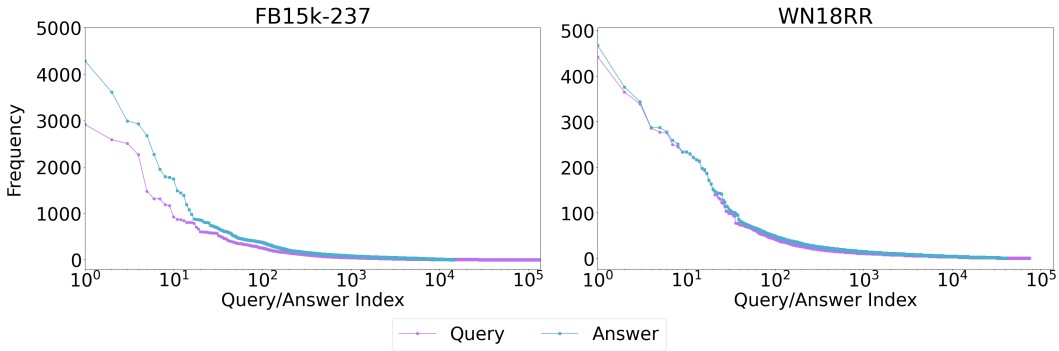

Figure 1: Appearance frequencies of queries and answers in the training data of FB15k-237 and WN18RR. Note that the indices are sorted from high frequency to low.

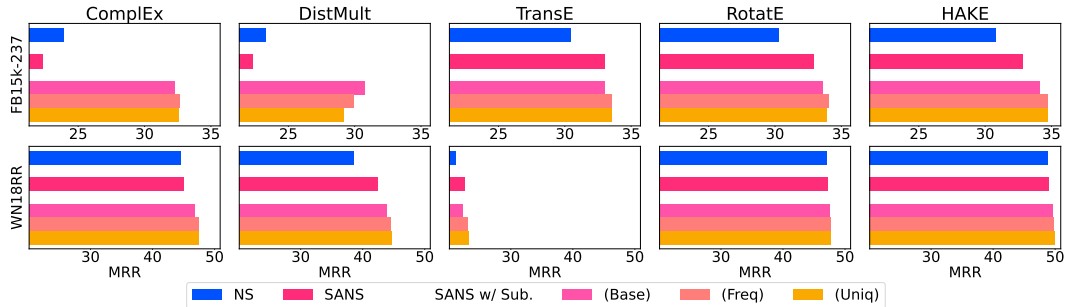

Figure 2: Performances of KGE models HAKE, RotatE, TransE, ComplEx, and DistMult on datasets FB15k-237, WN18RR using NS, SANS, and subsampling methods (noted as *Base, Freq, Uniq*).

when the training proceeds, the SANS can extract the model's potential compared with the NS loss with uniform distribution.

One of the left problems of KGE is the sparsity of KGs. Figure 1 shows the appearance frequency of queries and answers (entities) in the training data of FB15k-237 and WN18RR datasets. From the long-tail distribution of this figure, we can understand that both queries and answers necessary for training KGE models may suffer from the sparsity problem.

As a solution, several smoothing methods are used in KGE. Sun et al. (2019) import subsampling from word2vec (Mikolov et al., 2013) to KGE. Subsampling can smooth the appearance frequency of triplets and queries in KGs. Kamigaito & Hayashi (2022b) show a general formulation that covers the basic subsampling of Sun et al. (2019) (Base), their frequency-based subsampling (Freq) and unique-based subsampling (Uniq) for KGE. Kamigaito & Hayashi (2021) indicate that SANS has a similar effect of using label-smoothing (Szegedy et al., 2016) and thus SANS can smooth the frequencies of answers in training. Figure 2 shows the effectiveness of SANS and subsampling in KGC performance. From the figure, since FB15k-237 is more sparse (imbalanced) than WN18RR based on Figure 1, we can understand that difference in smoothing methods have more considerable influences than in models when target data is sparse.

While SANS and subsampling can improve model performance by smoothing the appearance frequencies of triplets, queries, and answers, their theoretical relationship is not clear, leaving their capabilities and deficiencies a question. For example, conventional works (Sun et al., 2019; Zhang et al., 2020b; Kamigaito & Hayashi, 2022b)[1] jointly use SANS and subsampling with no theoretical background. Thus, there is room for further performance improvement.

---

[1]Note that Sun et al. (2019); Zhang et al. (2020b) use subsampling in their released implementation without referring to it in their paper.

To solve the above problem, we theoretically and empirically study the difference of SANS and subsampling[2] on three common datasets and five popular KGE models. Our contributions are:

- By focusing on the smoothing targets, we theoretically reveal the difference between SANS and subsampling and induce a new NS loss, Triplet-based SANS (T-SANS), that can cover the smoothing target of both SANS and subsampling.
- We theoretically show that T-SANS with subsampling can potentially cover the conventional usages of SANS and subsampling.
- We empirically verify that T-SANS improves KGC performance on sparse KGs in terms of MRR.
- We empirically verify that T-SANS with subsampling can cover the conventional usages of SANS and subsampling in terms of MRR.

## 2 BACKGROUND

In this section, we describe the problem formulation for solving KGC by KGE and explain the conventional NS loss functions in KGE.

### 2.1 FORMULATION OF KGE

KGC is a research topic for automatically inferring new links in a KG that are likely but not yet known to be true. To infer the new links by KGE, we decompose KGs into a set of triplets (links). By using entities $e_i, e_j$ and their relation $r_k$, we represent the triplet as $(e_i, r_k, e_j)$. In a typical KGC task, a KGE model receives a query $(e_i, r_k, ?)$ or $(?, r_k, e_j)$ and predicts the entity corresponding to ? as an answer.

In KGE, a KGE model scores a triplet $(e_i, r_k, e_j)$ by using a scoring function $s_\theta(x, y)$, where $\theta$ denotes model parameters. Here, using a softmax function, we represent the existence probability $p_\theta(y|x)$ for an answer $y$ of the query $x$ as follows:

$$p_\theta(y|x) = \frac{\exp(s_\theta(x, y))}{\sum_{y' \in Y} \exp(s_\theta(x, y'))}, \tag{1}$$

where Y is a set of entities.

### 2.2 NS LOSS IN KGE

To train $s_\theta(x, y)$, we need to calculate losses for the observables $D = \{(x_1, y_1), \cdots, (x_n, y_n)\}$ that follow $p_d(x, y)$. Even if we can represent KGC by Eq. (1), it does not mean we can tractably perform KGC due to the large number of Y in KGs. For the reason of the computational cost, the NS loss (Mikolov et al., 2013) is used to approximate Eq. (1) by sampling false answers.

By modifying that of Mikolov et al. (2013), the following NS loss (Sun et al., 2019; Ahrabian et al., 2020) is commonly used in KGE:

$$\ell_{\text{NS}}(\theta) = -\frac{1}{|D|} \sum_{(x,y) \in D} \left[ \log(\sigma(s_\theta(x, y) + \tau)) + \frac{1}{\nu} \sum_{y_i \sim U}^{\nu} \log(\sigma(-s_\theta(x, y_i) - \tau)) \right], \tag{2}$$

where $U$ is the noise distribution that follows uniform distribution, $\sigma$ is the sigmoid function, $\nu$ is the number of negative samples per positive sample $(x, y)$, and $\tau$ is a margin term to adjust the value range decided by $s_\theta(x, y)$.

### 2.3 SMOOTHING METHODS FOR THE NS LOSS IN KGE

As shown in Figure 1, KGC needs to deal with the sparsity problem caused by low frequent queries and answers in KGs. Imposing smoothing on the appearance frequencies of queries and answers can mitigate this problem. The following subsections introduce subsampling (Mikolov et al., 2013; Sun et al., 2019; Kamigaito & Hayashi, 2022b) and SANS (Sun et al., 2019), the conventional smoothing methods for the NS loss in KGE.

---

[2]Our code and data will be available at `https://github.com/[innominated]`.

### 2.3.1 SUBSAMPLING

Subsampling (Mikolov et al., 2013) is a method to smooth the frequency of triplets or queries in the NS loss. Sun et al. (2019) import this approach from word embedding to KGE. Kamigaito & Hayashi (2022a;b) add some variants to subsampling for KGC and theoretically provide a unified expression of them as follows:

$$\ell_{\text{SUB}}(\theta)$$
$$= -\frac{1}{|D|} \sum_{(x,y)\in D} \left[ A(x,y;\alpha) \log(\sigma(s_\theta(x,y)+\tau)) + \frac{1}{\nu} \sum_{y_i \sim U}^{\nu} B(x,y;\alpha) \log(\sigma(-s_\theta(x,y_i)-\tau)) \right], \quad (3)$$

where $\alpha$ is a temperature term to adjust the frequecy of triplets and queries. Note that we incorporate $\alpha$ into Eq. (3) to consider various loss functions even though Kamigaito & Hayashi (2022a;b) do not consider $\alpha$. In this formulation, we can consider several assumptions for deciding $A(x,y;\alpha)$ and $B(x,y;\alpha)$. We introduce these assumptions in the following paragraphs:

**Base** As a basic subsampling approach, Sun et al. (2019) import the one originally used in word2vec Mikolov et al. (2013) to KGE learning, defined as follows:

$$A(x,y;\alpha) = B(x,y;\alpha) = \frac{\#(x,y)^{-\alpha}|D|}{\sum_{(x',y')\in D} \#(x',y')^{-\alpha}}, \quad (4)$$

where $\#$ is the symbol for frequency and $\#(x,y)$ represents the frequency of $(x,y)$. In word2vec, subsampling randomly discards a word by a probability $1 - \sqrt{t/f}$, where $t$ is a constant value and $f$ is a frequency of a word. This is similar to randomly keeping a word with a probability $\sqrt{t/f}$. Thus, we can understand that Eq. (4) follows the original use in word2vec. Since the actual $(x,y)$ occurs at most once in KGs, when $(x,y) = (e_i, r_k, e_j)$, they approximate the frequency of $(x,y)$ as:

$$\#(x,y) \approx \#(e_i, r_k) + \#(r_k, e_j), \quad (5)$$

based on the approximation of n-gram language modeling (Katz, 1987).

**Freq** Kamigaito & Hayashi (2022b) propose frequency-based subsamping (Freq) by assuming a case that $(x,y)$ originally has a frequency, but the observed one in the KG is at most 1.

$$A(x,y;\alpha) = \frac{\#(x,y)^{-\alpha}|D|}{\sum_{(x',y')\in D} \#(x',y')^{-\alpha}}, \quad B(x,y;\alpha) = \frac{\#x^{-\alpha}|D|}{\sum_{x'\in D} \#x'^{-\alpha}}. \quad (6)$$

**Uniq** Kamigaito & Hayashi (2022b) also propose unique-based subsamping (Uniq) by assuming a case that the originally frequency and the observed one in the KG are both 1.

$$A(x,y;\alpha) = B(x,y;\alpha) = \frac{\#x^{-\alpha}|D|}{\sum_{x'\in D} \#x'^{-\alpha}}. \quad (7)$$

### 2.3.2 SANS LOSS

SANS is originally proposed as a kind of NS loss to train KGE models efficiently by considering negative samples close to their corresponding positive ones. Kamigaito & Hayashi (2021) show that using SANS is similar to imposing label-smoothing on Eq. (1). Thus, SANS is a method to smooth the frequency of answers in the NS loss. The SANS loss is represented as follows:

$$\ell_{\text{SANS}}(\theta) = -\frac{1}{|D|} \sum_{(x,y)\in D} \left[ \log(\sigma(s_\theta(x,y)+\tau)) + \sum_{y_i \sim U}^{\nu} p_\theta(y_i|x;\beta) \log(\sigma(-s_\theta(x,y_i)-\tau)) \right], \quad (8)$$

$$p_\theta(y_i|x;\beta) \approx \frac{\exp(\beta s_\theta(x,y_i))}{\sum_{j=1}^{\nu} \exp(\beta s_\theta(x,y_j))}, \quad (9)$$

where $\beta$ is a temperature to adjust the distribution of negative sampling. Different from subsampling, SANS uses $p_\theta(y_i|x;\beta)$ that is predicted by a model $\theta$ to adjust the frequency of the answer $y_i$. Since $p_\theta(y_i|x;\beta)$ is essentially a noise distribution, it does not receive any gradient during training.

Table 1: The characteristics of each smoothing method for the NS loss in KGE (See §2.3 for the details.) and our proposed T-SANS. ✓ and △ respectively denote the method smooths the probability directly and indirectly. × denotes the method does not smooth the probability.

| Method | | Smoothing | | | Remarks |
|---|---|---|---|---|---|
| | | $p(x,y)$ | $p(y\|x)$ | $p(x)$ | |
| Subsampling | Base | ✓ | △ | △ | $p(y\|x)$ and $p(x)$ are influenced by $p(x,y)$. |
| | Uniq | △ | × | ✓ | $p(x,y)$ is indirectly controlled by $p(x)$. |
| | Freq | ✓ | △ | ✓ | $p(y\|x)$ is indirectly controlled by $p(x,y)$ or $p(x)$. |
| SANS | | △ | ✓ | × | $p(x,y)$ is indirectly controlled by $p(y\|x)$. |
| T-SANS | | ✓ | ✓ | ✓ | |

## 3 TRIPLET-BASED SANS

In this section, we explain our proposed Triplet-based SANS (T-SANS) in detail. We first show the overview of our T-SANS through the comparison with the conventional smoothing methods of the NS loss for KGE (See §2.3) in §3.1 and after that we explain the details of T-SANS through its mathematical formulations in §3.2 and §3.3.

### 3.1 OVERVIEW

T-SANS is fundamentally different from SANS, with SANS only taking into account the conditional probability of negative samples and T-SANS being a loss function that considers the joint probability of the pair of queries and their answers.

Table 1 shows the characteristics of T-SANS and the conventional smoothing methods of the NS loss for KGE introduced in §2.3. These characteristics are based on the decomposition of $p_d(x, y)$, the appearance probability for the triplet $(x, y)$, into that of its answer $p_d(y|x)$ and query $p(x)$:

$$p_d(x, y) = p_d(y|x)p_d(x) \qquad (10)$$

In Eq. (10), smoothing both $p_d(y|x)$ and $p_d(x)$ is similar to smoothing $p_d(x, y)$. However, smoothing $p_d(x, y)$ does not ensure smoothing both $p_d(x)$ and $p_d(y|x)$ considering the case of only one of them being smoothed, and the left one being still sparse. Similarly, smoothing only $p_d(x)$ or $p_d(y|x)$ does not ensure $p_d(x, y)$ being smoothed due to the case where one of them is still sparse. In Table 1, we denote such case where the method can influence the probability, but no guarantee of the probability be smoothed as △.

In T-SANS, we target to smooth $p_d(x, y)$ by smoothing both $p_d(y|x)$ and $p_d(x)$ based on Eq. (10).

### 3.2 FORMULATION

Here, we induce T-SANS from SANS with targeting to smooth $p_d(x, y)$ by smoothing both $p_d(y|x)$ and $p_d(x)$. First, we assume a simple replacement from $p_\theta(y|x)$ to $p_\theta(x, y)$ in $\ell_{\text{SANS}}(\theta)$ of Eq. (9):

$$-\frac{1}{|D|} \sum_{(x,y)\in D} \left[ \log(\sigma(s_\theta(x, y) + \tau)) + \sum_{y_i \sim U}^{\nu} p_\theta(x, y_i) \log(\sigma(-s_\theta(x, y_i) - \tau)) \right]. \qquad (11)$$

However, using Eq. (11) causes an imbalanced loss between the first and second terms since the sum of $p_\theta(x, y_i)$ on all negative samples is not always 1. Thus, Eq. (11) is impractical as a loss function.

As a solution, we focus on the decomposition $p_\theta(x, y) = p_\theta(y|x)p_\theta(x)$ and the fact that the sum of $p_\theta(y|x)$ of all negative samples is always 1. By using $p_\theta(x)$ to make a balance between the first and

Table 2: The relationship between the loss functions from the viewpoint of the unified NS loss, $\ell_{\mathrm{UNI}}(\theta)$ in Eq. (16).

| Temperature | | | Induced NS Loss |
|---|---|---|---|
| $\alpha$ | $\beta$ | $\gamma$ | |
| $= 0$ | $= 0$ | $= 0$ | Equivalent to $\ell_{\mathrm{NS}}(\theta)$, the basic NS loss in KGE (Eq. (2)) |
| $= 0$ | $= 0$ | $\neq 0$ | Currently does not exist |
| $= 0$ | $\neq 0$ | $= 0$ | Proportional to $\ell_{\mathrm{SANS}}(\theta)$, the SANS loss (Eq. (9)) |
| $= 0$ | $\neq 0$ | $\neq 0$ | Equivalent to our $\ell_{\mathrm{T\text{-}SANS}}(\theta)$, the T-SANS loss (Eq. (12)) |
| $\neq 0$ | $= 0$ | $= 0$ | Proportional to $\ell_{\mathrm{NS}}(\theta)$, the basic NS loss in KGE (Eq. (2)) with subsampling in §2.3 |
| $\neq 0$ | $= 0$ | $\neq 0$ | Currently does not exist |
| $\neq 0$ | $\neq 0$ | $= 0$ | Proportional to $\ell_{\mathrm{SANS}}(\theta)$, the SANS loss (Eq. (9)) with subsampling in §2.3 |
| $\neq 0$ | $\neq 0$ | $\neq 0$ | Equivalent to our $\ell_{\mathrm{UNI}}(\theta)$, the unified NS loss in KGE (Eq. (16)) and also equivalent to our $\ell_{\mathrm{T\text{-}SANS}}(\theta)$, the T-SANS loss (Eq. (12)) with subsampling in §2.3 |

second loss term, we can modify Eq. (11) and induce our T-SANS as follows:

$$\ell_{\mathrm{T\text{-}SANS}}(\theta)$$
$$= -\frac{1}{|D|} \sum_{(x,y) \in D} p_\theta(x; \gamma) \Big[ \log(\sigma(s_\theta(x, y) + \tau)) + \sum_{y_i \sim U}^{\nu} p_\theta(y_i | x; \beta) \log(\sigma(-s_\theta(x, y_i) - \tau)) \Big], \quad (12)$$

$$p_\theta(x; \gamma) = \sum_{y_i \in D} p_\theta(x, y_i; \gamma), \quad p_\theta(x, y_i; \gamma) = \frac{\exp(\gamma s_\theta(x, y_i))}{\sum_{(x', y') \in D} \exp(\gamma s_\theta(x', y'))}, \quad (13)$$

where $\gamma$ is a temperature term to smooth the frequency of queries. Since T-SANS uses a noise distribution decided by $p_\theta(x; \gamma)$ and $p_\theta(y_i | x; \beta)$, it does not propagate gradients through probabilities for negative samples, and thus, memory usage is not increased.

### 3.3 THEORETICAL INTERPRETATION

In this subsection, we discuss the difference and similarities between T-SANS and other smoothing methods for the NS loss in KGE. As shown in Table 1, the subsampling methods, Base and Freq, can smooth triplet frequencies similar to our T-SANS. To investigate T-SANS from the view point of subsampling, we reformulate Eq. (12) as follows:

$$\ell_{\mathrm{T\text{-}SANS}}(\theta)$$
$$= -\frac{1}{|D|} \sum_{(x,y) \in D} A(x, y; \gamma) \Big[ \log(\sigma(s_\theta(x, y) + \tau)) + \sum_{y_i \sim U}^{\nu} B(x, y; \beta, \gamma) \log(\sigma(-s_\theta(x, y_i) - \tau)) \Big], \quad (14)$$

$$A(x, y; \gamma) = p_\theta(x; \gamma), \quad B(x, y; \beta, \gamma) = p_\theta(y_i | x; \beta) p_\theta(x; \gamma). \quad (15)$$

Apart from the temperature terms, $\alpha$, $\beta$, and $\gamma$, we can see that the general formulation of subsampling in Eq. (3) and the above Eq. (14) has the same formulation. Thus, T-SANS is not merely an extension of SANS but also a novel subsampling method.

Even though their similar characteristic, T-SANS and subsampling have an essential difference: T-SANS smooths the frequencies by model-predicted distributions as in Eq. (13), and the conventional subsampling methods smooth them by counting appearance frequencies on the observed data as in Eq. (4), (5), (6), and (7). For instance, our T-SANS can work even when the entity or relations included in the target triplet appear more than once, which is theoretically different from conventional approaches.

Since the superiority of using either model-based or count-based frequencies depends on the model and dataset, we empirically investigate this point through our experiments.

## 4 UNIFIED INTERPRETATION OF SANS AND SUBSAMPLING

In the previous section, we understand that our T-SANS can smooth triplets, queries, and answers partially covered by SANS and subsampling methods. On the other hand, T-SANS only relies

on model-predicted frequencies to smooth the frequencies. Neubig & Dyer (2016) point out the benefits of combining count-based and model-predicted frequencies in language modeling. This section integrates smoothing methods for the NS loss in KGE from a unified interpretation.

## 4.1 FORMULATION

We formulate the unified loss function by introducing subsampling into our T-SANS as follows:

$$\ell_{\text{UNI}}(\theta) = -\frac{1}{|D|} \sum_{(x,y)\in D} p_\theta(x;\gamma) \Big[ A(x,y;\alpha) \log(\sigma(s_\theta(x,y)+\tau))$$
$$+ \eta \sum_{y_i \sim U}^{\nu} B(x,y;\alpha) p_\theta(y_i|x;\beta) \log(\sigma(-s_\theta(x,y_i)-\tau)) \Big], \tag{16}$$

where $\eta$ is a hyperparamter that can be any value to absorb the difference between the three different subsampling methods, Base, Uniq, and Freq.

Here, we can induce the NS losses shown in our paper from Eq. (16) by changing the temperature parameters $\alpha$, $\beta$, and $\gamma$. Table 2 shows the induced loss functions from our $\ell_{\text{UNI}}(\theta)$. Note that since $p_\theta(x;\gamma)$ only appears in our T-SANS, canceling $p_\theta(x;\gamma)$ by $\gamma = 0$ induces not an equivalent but a proportional relationship to the conventional NS loss.

## 4.2 THEORETICAL INTERPRETATION

As shown in Table 2, T-SANS w/ subsampling has characteristics of all smoothing methods for the NS loss in KGE introduced in this paper. Therefore, we can expect higher performance of T-SANS w/ subsampling than the combination of conventional methods, the basic NS, SANS, and subsampling. However, because T-SANS w/ subsampling uses subsampling in §2.3, we need to choose the one from Base, Uniq, and Freq for T-SANS w/ subsampling. Since this part is out of the scope of our theoretical interpretation, we investigate this part in the experiments.

## 5 EXPERIMENTS

In this section, we investigate our theoretical interpretation in §3.3 and §4.2 through experiments.

## 5.1 EXPERIMENTAL SETTINGS

**Datasets** We used three common datasets, FB15k-237 (Toutanova & Chen, 2015), WN18RR, and YAGO3-10 (Dettmers et al., 2018).[3]

**Comparison Methods** As comparison methods, we used ComplEx (Trouillon et al., 2016), DistMult (Yang et al., 2015), TransE (Bordes et al., 2013), RotatE (Sun et al., 2019), and HAKE (Zhang et al., 2020a). We followed the original settings of Sun et al. (2019) for ComplEx, DistMult, TransE, and RotatE with their implementation[4] and the original settings of Zhang et al. (2020a) for HAKE with their implementation[5]. We tuned temperature $\gamma$ on the validation split for each dataset.

**Metrics** We employed conventional metrics in KGC, i.e., MRR, Hits@1 (H@1), Hits@3 (H@3), and Hits@10 (H@10) and reported the average scores and their standard deviations by three different runs with fixed random seeds.

## 5.2 RESULTS

The full experimental results are listed in Appendix B, including Table 4, 5, and 6 of Appendix B.1, and training loss curves and validation MRR curves for each smoothing method in Figure 5, 6, and 7 of Appendix B.2. Since these tables are large, we discuss them individually, focusing on important information in the following subsections.

---

[3]Table 3 in Appendix A shows the dataset statistics for each dataset.
[4]`https://github.com/DeepGraphLearning/KnowledgeGraphEmbedding`
[5]`https://github.com/MIRALab-USTC/KGE-HAKE`

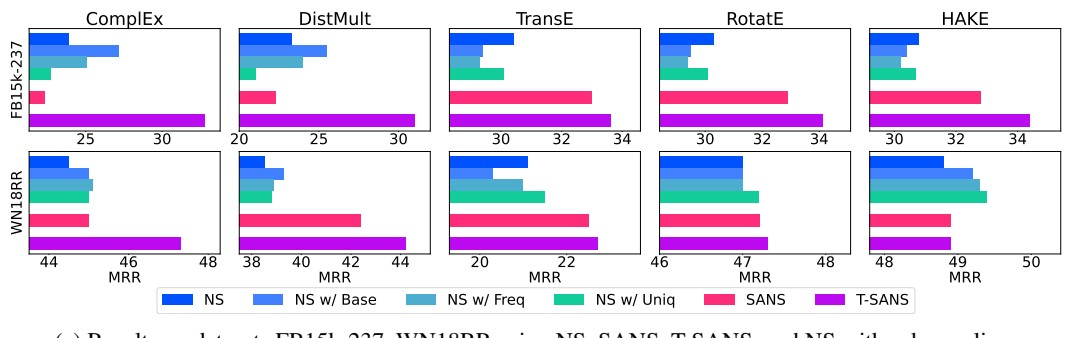

(a) Results on datasets FB15k-237, WN18RR using NS, SANS, T-SANS, and NS with subsampling.

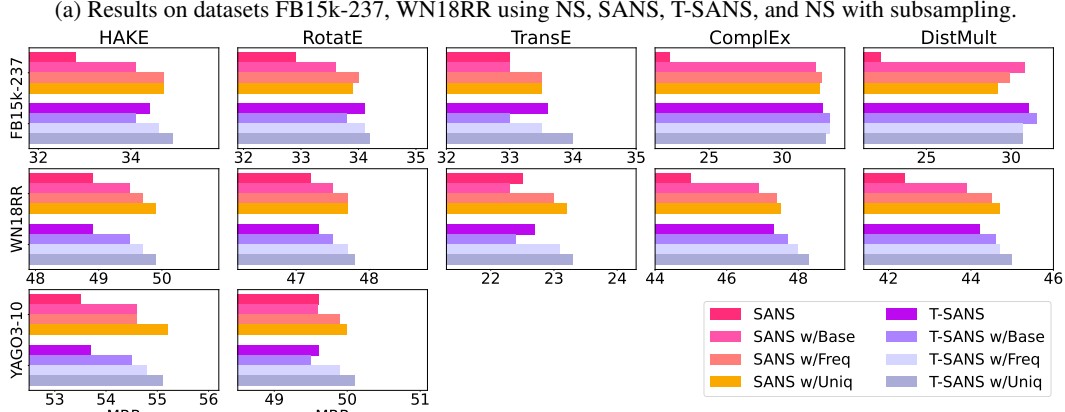

(b) Results on datasets FB15k-237, WN18RR, YAGO3-10 using SANS, T-SANS, and those with subsampling.

Figure 3: KGC performance on common KGs (Notations are the same as in Figure 2).

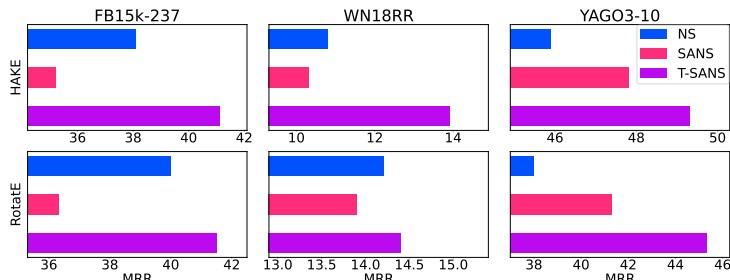

Figure 4: KGC performance on artificially created imbalanced KGs derived from common KGs.

### 5.2.1 EFFECTIVENESS OF T-SANS

Figure 3a shows the MRR scores of each method. From the result, we can understand the effectiveness of considering triplet information in SANS as conducted in T-SANS. Thus, the result is along with our expectation in §3.3 that T-SANS can cover the role of subsampling methods. However, as the result of HAKE in WN18RR shows, there is a case that subsampling methods outperform T-SANS. As discussed in §3.3, using only T-SANS does not cover all combinations of NS loss and subsampling. Considering this theoretical fact, we further compare T-SANS with subsampling and the NS loss with subsampling.

### 5.2.2 VALIDITY OF THE UNIFIED INTERPRETATION

Figure 3b shows the result for each configuration. We can see performance improvements by using subsampling in both SANS and T-SANS. Furthermore, in almost all cases, T-SANS with subsampling achieve the highest MRR. This observation is along with the theoretical conclusion in §3.3 that T-SANS with subsampling can cover the characteristic of other NS loss in terms of smoothing.

On the other hand, the results of HAKE on YAGO3-10 show the different tendency that SANS with subsampling achieves the best MRR instead of T-SANS. Because the model prediction estimates the triplet frequencies, T-SANS is influenced by the selected model. Therefore, carefully choosing the combination of a loss function and model is still effective in improving KGC performance on the NS loss with subsampling.

# 6 ANALYSIS

We analyze how T-SANS mitigate the sparsity problem in imbalanced KGs commonly caused by low frequent triplets in KGC. By considering that all triplets in KGs appear at most once, we focus on query frequencies. We extracted triplets with the highest or lowest 5% frequent queries in training, validation, and test splits as artificial data for the investigation. Note that we show their detailed statistics in Table 7 of Appendix C.1.

Figure 4 shows MRRs for each model on each extremely sparse dataset. From the result, we can understand that T-SANS can perform even much better in KGC when KGs are extremely sparse and imbalanced. You can see further details of the result in Table 7 of Appendix C.2.

# 7 RELATED WORK

Mikolov et al. (2013) initially propose the NS loss of the frequent words to train their word embedding model, word2vec. Trouillon et al. (2016) introduce the NS loss to KGE to speed up training. Melamud et al. (2017) use the NS loss to train the language model. In contextualized pre-trained embeddings, Clark et al. (2020a) indicate that a BERTDevlin et al. (2019)-like model ELECTRA Clark et al. (2020b) uses the NS loss to perform better and faster than language models.

Sun et al. (2019) extend the NS loss to SANS loss for KGE and proposed their noise distribution, which is subsampled by a uniformed probability $p_\theta(y_i|x)$. Kamigaito & Hayashi (2021) point out the sparseness problem of KGs through their theoretical analysis of the NS loss in KGE. Furthermore, Kamigaito & Hayashi (2022b) reveal that subsampling Mikolov et al. (2013) can alleviate the sparseness problem in the NS for KGE and conclude three assumptions for subsampling, Base, Freq, and Uniq.

Through our work, we theoretically clarify the position of the previous works on SANS loss and subsampling from the viewpoint of smoothing methods for the NS loss in KGE. Since our work unitedly interprets SANS loss and subsampling, our proposed T-SANS inherits the advantages of conventional works and can deal with the sparsity problem in the NS loss for KGE.

# 8 CONCLUSION

We reveal the relationships between SANS loss and subsampling for the KG completion task through theoretical analysis. We explain that SANS loss and subsampling under three assumptions, Base, Freq, and Uniq have similar roles to mitigate the sparseness problem of queries and answers of KGs by smoothing the frequencies of queries and answers. Furthermore, based on our interpretation, we induce a new loss function, Triplet-based SANS (T-SANS), by integrating SANS loss and subsampling. We also introduce a theoretical interpretation that T-SANS with subsampling can cover all conventional combinations of SANS loss and subsampling.

We verified our interpretation by empirical experiments in three common datasets, FB15k-237, WN18RR, and YAGO3-10, and five popular KGE models, ComplEx, DistMult, TransE, RotatE, and HAKE. The experimental results show that our T-SANS loss can outperform subsampling and SANS loss with many models in terms of MRR as expected by our theoretical interpretation. Furthermore, the combinatorial use of T-SANS and subsampling achieved comparable or better performance than other combinations and showed the validity of our theoretical interpretation that T-SANS with subsampling can cover all conventional combinations of SANS loss and subsampling in KGE.

In our future work, we plan to generalize T-SANS for word embeddings and item recommendations tasks, since these are similar to the special case of KGs whose triplets have the same relationships.

ETHICS STATEMENT

We used the publicly available datasets, FB15k-237, WN18RR, and YAGO3-10, to train and evaluate KGE models, and there is no ethical consideration.

REPRODUCIBILITY STATEMENT

We used the publicly available code to implement KGE models, ComplEx, DistMult, TransE, RotatE, and HAKE with the author-provided hyperparameters as described in §5.1. Regarding the temperature parameter $\gamma$, we tuned it on the validation split for each dataset and reported the values in Table 4, 5, and 6 of Appendix B. Our code and data will be available at `https://github.com/[innominated]`.

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

Table 3: Statistics for each dataset.

| Dataset | Split | Tuple | Query | Entity | Relation |
|---|---|---|---|---|---|
| FB15k-237 | Total | 310,116 | 150,508 | 14,541 | 237 |
| | #Train | 272,115 | 138,694 | 14,505 | 237 |
| | #Valid | 17,535 | 19,750 | 9,809 | 223 |
| | #Test | 20,466 | 22,379 | 10,348 | 224 |
| WN18RR | Total | 93,003 | 77,479 | 40,943 | 11 |
| | #Train | 86,835 | 74,587 | 40,559 | 11 |
| | #Valid | 3,034 | 5,431 | 5,173 | 11 |
| | #Test | 3,134 | 5,565 | 5,323 | 11 |
| YAGO3-10 | Total | 1,089,040 | 372,775 | 123,182 | 37 |
| | #Train | 1,079,040 | 371,077 | 123,143 | 37 |
| | #Valid | 5,000 | 8,534 | 7,948 | 33 |
| | #Test | 5,000 | 8,531 | 7,937 | 34 |

## A    DATASET STATISTICS

Table 3 shows the dataset statistics for each dataset introduced in §5.1.

## B    FULL EXPERIMENTAL RESULTS

### B.1    RESULTS TABLES

Table 4, 5, and 6 list all results in our experiments explained in §5.2. In these tables, the bold scores are the best results for each subsampling type (e.g. *None*, *Base*, *Freq*, and *Uniq.*), † indicates the best scores for each model, *SD* denotes the standard deviation of the three trials, and $\gamma$ denotes the temperature chosen by development data.

### B.2    TRAINING LOSS AND VALIDATION MRR CURVE

From these figures, we can understand that the convergence of T-SANS loss is as well as SANS and NS loss on datasets FB15k-237, WN18RR, and YAGO3-10 for each KGE model. Meanwhile, the time complexity of T-SANS is the same with SANS and NS loss too.

## C    SPARSE QUERIES

### C.1    DATA STATISTICS

Table 7 shows detailed statistics of the created sparse data explained in §6.

### C.2    DETAILED RESULTS

Table 8 shows the detailed results on the created sparse data expained in §6.

Table 4: Results on FB15k-237.

| Model | Subsampling Assumption | Subsampling Loss | MRR Mean | MRR SD | H@1 Mean | H@1 SD | H@3 Mean | H@3 SD | H@10 Mean | H@10 SD | $\gamma$ |
|---|---|---|---|---|---|---|---|---|---|---|---|
| | | | FB15k-237 | | | | | | | | |
| ComplEx | None | NS | 23.9 | 0.2 | 15.8 | 0.1 | 26.1 | 0.3 | 40.0 | 0.2 | - |
| | | SANS | 22.3 | 0.1 | 13.8 | 0.1 | 24.2 | 0.0 | 39.5 | 0.2 | - |
| | | T-SANS | **32.8** | 0.2 | **23.2** | 0.1 | **36.2** | 0.2 | **52.2** | 0.1 | -2 |
| | Base | NS | 27.2 | 0.1 | 19.1 | 0.1 | 29.5 | 0.1 | 43.0 | 0.2 | - |
| | | SANS | 32.3 | 0.0 | 23.0 | 0.1 | 35.4 | 0.1 | 51.2 | 0.1 | - |
| | | T-SANS | †**33.3** | 0.0 | †**23.8** | 0.1 | †**36.9** | 0.1 | †**52.7** | 0.0 | -1 |
| | Freq | NS | 25.1 | 0.2 | 17.1 | 0.3 | 27.4 | 0.2 | 41.0 | 0.2 | - |
| | | SANS | 32.7 | 0.1 | 23.6 | 0.1 | 36.0 | 0.1 | 51.2 | 0.1 | - |
| | | T-SANS | †**33.3** | 0.0 | †**23.8** | 0.0 | 36.8 | 0.1 | 52.1 | 0.2 | -0.5 |
| | Uniq | NS | 22.8 | 0.4 | 14.7 | 0.5 | 24.7 | 0.4 | 39.0 | 0.1 | - |
| | | SANS | 32.6 | 0.0 | **23.5** | 0.1 | 35.8 | 0.1 | 51.2 | 0.1 | - |
| | | T-SANS | **33.0** | 0.1 | **23.5** | 0.1 | **36.5** | 0.1 | **52.1** | 0.1 | -0.5 |
| DistMult | None | NS | 23.3 | 0.1 | 15.6 | 0.1 | 25.7 | 0.1 | 38.4 | 0.1 | - |
| | | SANS | 22.3 | 0.1 | 14.0 | 0.2 | 24.1 | 0.1 | 39.2 | 0.0 | - |
| | | T-SANS | **31.0** | 0.1 | **21.7** | 0.1 | **34.0** | 0.1 | **49.6** | 0.1 | -1 |
| | Base | NS | 25.4 | 0.1 | 17.9 | 0.1 | 27.6 | 0.1 | 40.4 | 0.1 | - |
| | | SANS | 30.8 | 0.1 | 21.9 | 0.1 | 33.6 | 0.1 | 48.4 | 0.1 | - |
| | | T-SANS | †**31.5** | 0.1 | †**22.4** | 0.1 | †**34.6** | 0.1 | †**49.7** | 0.0 | -0.5 |
| | Freq | NS | 24.0 | 0.1 | 16.7 | 0.2 | 25.9 | 0.1 | 38.4 | 0.1 | - |
| | | SANS | 29.9 | 0.0 | 21.2 | 0.1 | 32.8 | 0.0 | 47.5 | 0.1 | - |
| | | T-SANS | **30.7** | 0.0 | 21.6 | 0.0 | **34.0** | 0.0 | **49.0** | 0.0 | -1 |
| | Uniq | NS | 21.0 | 0.1 | 13.5 | 0.2 | 22.8 | 0.2 | 36.3 | 0.2 | - |
| | | SANS | 29.2 | 0.0 | 20.5 | 0.1 | 31.9 | 0.0 | 46.7 | 0.0 | - |
| | | T-SANS | **30.7** | 0.1 | **21.5** | 0.1 | **33.8** | 0.1 | **49.3** | 0.1 | -2 |
| TransE | None | NS | 30.4 | 0.0 | 21.3 | 0.1 | 33.4 | 0.1 | 48.5 | 0.0 | - |
| | | SANS | 33.0 | 0.1 | 22.9 | 0.1 | 37.2 | 0.1 | †**53.0** | 0.1 | - |
| | | T-SANS | **33.6** | 0.0 | **23.9** | 0.0 | **37.3** | 0.0 | †**53.0** | 0.1 | -0.5 |
| | Base | NS | 29.4 | 0.1 | 20.0 | 0.1 | 32.8 | 0.0 | 48.1 | 0.0 | - |
| | | SANS | **33.0** | 0.1 | 23.1 | 0.1 | **36.8** | 0.1 | **52.7** | 0.1 | - |
| | | T-SANS | **33.0** | 0.0 | 23.1 | 0.0 | **36.8** | 0.1 | **52.7** | 0.1 | -0.1 |
| | Freq | NS | 29.3 | 0.1 | 20.0 | 0.1 | 32.8 | 0.1 | 47.8 | 0.1 | - |
| | | SANS | **33.5** | 0.0 | **23.9** | 0.1 | **37.2** | 0.1 | **52.8** | 0.1 | - |
| | | T-SANS | **33.5** | 0.1 | **23.9** | 0.1 | **37.2** | 0.0 | **52.8** | 0.1 | -0.1 |
| | Uniq | NS | 30.1 | 0.1 | 21.0 | 0.1 | 33.6 | 0.0 | 48.0 | 0.0 | - |
| | | SANS | 33.5 | 0.0 | 23.9 | 0.0 | 37.3 | 0.2 | 52.7 | 0.1 | - |
| | | T-SANS | †**34.0** | 0.1 | †**24.5** | 0.1 | †**37.7** | 0.1 | †**53.0** | 0.1 | 0.5 |
| RotatE | None | NS | 30.3 | 0.0 | 21.4 | 0.1 | 33.2 | 0.1 | 48.4 | 0.1 | - |
| | | SANS | 32.9 | 0.1 | 22.8 | 0.1 | 36.8 | 0.0 | 53.1 | 0.2 | - |
| | | T-SANS | **34.1** | 0.1 | **24.6** | 0.1 | **37.7** | 0.1 | †**53.3** | 0.1 | -0.5 |
| | Base | NS | 29.5 | 0.0 | 20.3 | 0.0 | 32.7 | 0.1 | 47.9 | 0.0 | - |
| | | SANS | 33.6 | 0.1 | 23.9 | 0.1 | 37.3 | 0.1 | **53.1** | 0.1 | - |
| | | T-SANS | **33.8** | 0.0 | **24.2** | 0.0 | **37.4** | 0.0 | 53.0 | 0.1 | -0.5 |
| | Freq | NS | 29.4 | 0.1 | 20.2 | 0.1 | 32.6 | 0.1 | 47.6 | 0.1 | - |
| | | SANS | 34.0 | 0.1 | **24.6** | 0.0 | **37.7** | 0.0 | 53.0 | 0.0 | - |
| | | T-SANS | **34.1** | 0.0 | **24.6** | 0.0 | **37.7** | 0.0 | **53.1** | 0.1 | -0.01 |
| | Uniq | NS | 30.1 | 0.0 | 21.2 | 0.1 | 33.3 | 0.1 | 47.7 | 0.1 | - |
| | | SANS | 33.9 | 0.1 | 24.4 | 0.1 | 37.6 | 0.1 | 52.9 | 0.1 | - |
| | | T-SANS | †**34.2** | 0.0 | †**24.7** | 0.1 | †**37.8** | 0.0 | **53.1** | 0.1 | 0.5 |
| HAKE | None | NS | 30.8 | 0.1 | 21.8 | 0.1 | 33.8 | 0.1 | 48.6 | 0.1 | - |
| | | SANS | 32.8 | 0.2 | 22.7 | 0.3 | 36.9 | 0.1 | 52.8 | 0.1 | - |
| | | T-SANS | **34.4** | 0.1 | **24.9** | 0.1 | **37.9** | 0.2 | **53.6** | 0.0 | -0.5 |
| | Base | NS | 30.4 | 0.1 | 21.6 | 0.1 | 33.3 | 0.1 | 48.2 | 0.0 | - |
| | | SANS | **34.1** | 0.1 | **24.4** | 0.1 | **37.9** | 0.1 | 53.6 | 0.2 | - |
| | | T-SANS | **34.1** | 0.0 | **24.4** | 0.0 | **37.9** | 0.0 | **53.7** | 0.0 | -0.05 |
| | Freq | NS | 30.2 | 0.1 | 21.5 | 0.0 | 33.1 | 0.0 | 47.7 | 0.1 | - |
| | | SANS | **34.7** | 0.0 | **25.2** | 0.1 | **38.2** | 0.0 | **53.8** | 0.1 | - |
| | | T-SANS | 34.6 | 0.0 | 25.0 | 0.1 | **38.2** | 0.2 | 53.7 | 0.1 | 0.05 |
| | Uniq | NS | 30.7 | 0.1 | 22.2 | 0.1 | 33.5 | 0.1 | 48.0 | 0.1 | - |
| | | SANS | 34.7 | 0.1 | 25.1 | 0.1 | 38.3 | 0.1 | 53.9 | 0.1 | - |
| | | T-SANS | †**34.9** | 0.0 | †**25.4** | 0.0 | †**38.6** | 0.1 | †**54.0** | 0.1 | 0.5 |

Table 5: Results on WN18RR.

| Model | Subsampling | | MRR Mean | SD | H@1 Mean | SD | H@3 Mean | SD | H@10 Mean | SD | $\gamma$ |
|---|---|---|---|---|---|---|---|---|---|---|---|
| ComplEx | None | NS | 44.5 | 0.1 | 38.1 | 0.2 | 48.3 | 0.2 | 55.5 | 0.1 | - |
| | | SANS | 45.0 | 0.1 | 41.0 | 0.1 | 46.5 | 0.3 | 53.3 | 0.3 | - |
| | | T-SANS | **47.3** | 0.0 | **43.3** | 0.0 | **49.1** | 0.1 | **55.7** | 0.1 | -2 |
| | Base | NS | 45.0 | 0.1 | 38.9 | 0.1 | 48.6 | 0.2 | 55.7 | 0.1 | - |
| | | SANS | 46.9 | 0.1 | 42.7 | 0.2 | 48.5 | 0.2 | 55.5 | 0.2 | - |
| | | T-SANS | **47.7** | 0.2 | **43.6** | 0.1 | **49.3** | 0.2 | **55.9** | 0.3 | -2 |
| | Freq | NS | 45.1 | 0.1 | 38.9 | 0.1 | 48.8 | 0.2 | 56.0 | 0.2 | - |
| | | SANS | 47.4 | 0.1 | 43.2 | 0.1 | 49.2 | 0.2 | 56.0 | 0.2 | - |
| | | T-SANS | **48.0** | 0.1 | 43.9 | 0.1 | †**49.7** | 0.1 | **56.1** | 0.1 | -2 |
| | Uniq | NS | 45.0 | 0.1 | 38.7 | 0.1 | 48.8 | 0.1 | 56.0 | 0.3 | - |
| | | SANS | 47.5 | 0.1 | 43.3 | 0.1 | 49.1 | 0.2 | 56.2 | 0.2 | - |
| | | T-SANS | †**48.3** | 0.1 | †**44.4** | 0.2 | 49.6 | 0.1 | †**56.3** | 0.2 | -1 |
| DistMult | None | NS | 38.5 | 0.2 | 30.6 | 0.3 | 42.9 | 0.2 | 52.5 | 0.1 | - |
| | | SANS | 42.4 | 0.0 | 38.2 | 0.1 | 43.7 | 0.0 | 51.0 | 0.2 | - |
| | | T-SANS | **44.2** | 0.1 | **40.1** | 0.1 | **45.3** | 0.1 | **53.2** | 0.2 | -2 |
| | Base | NS | 39.3 | 0.2 | 31.9 | 0.2 | 43.3 | 0.1 | 53.0 | 0.2 | - |
| | | SANS | 43.9 | 0.1 | 39.4 | 0.1 | 45.2 | 0.1 | 53.3 | 0.2 | - |
| | | T-SANS | **44.6** | 0.0 | **40.5** | 0.2 | **45.7** | 0.1 | **53.9** | 0.1 | -2 |
| | Freq | NS | 39.0 | 0.2 | 31.2 | 0.2 | 43.2 | 0.1 | 52.9 | 0.2 | - |
| | | SANS | 44.5 | 0.1 | 40.0 | 0.1 | **46.0** | 0.1 | **54.2** | 0.2 | - |
| | | T-SANS | **44.7** | 0.1 | **40.5** | 0.2 | 45.8 | 0.1 | 54.0 | 0.2 | -2 |
| | Uniq | NS | 38.8 | 0.2 | 30.8 | 0.2 | 43.1 | 0.1 | 53.0 | 0.2 | - |
| | | SANS | 44.7 | 0.1 | 40.1 | 0.1 | †**46.2** | 0.3 | 54.3 | 0.0 | - |
| | | T-SANS | †**45.0** | 0.1 | †**40.7** | 0.1 | 46.1 | 0.2 | †**54.5** | 0.2 | -0.5 |
| TransE | None | NS | 21.1 | 0.0 | 2.1 | 0.1 | 36.5 | 0.2 | 50.4 | 0.2 | - |
| | | SANS | 22.5 | 0.1 | 1.7 | 0.1 | **40.2** | 0.1 | 52.5 | 0.2 | - |
| | | T-SANS | **22.7** | 0.0 | **2.5** | 0.0 | 39.5 | 0.2 | **53.4** | 0.1 | 0.5 |
| | Base | NS | 20.3 | 0.1 | **1.6** | 0.1 | 35.1 | 0.2 | 49.9 | 0.2 | - |
| | | SANS | 22.3 | 0.0 | 1.3 | 0.1 | **40.2** | 0.1 | 52.9 | 0.1 | - |
| | | T-SANS | **22.4** | 0.1 | 1.4 | 0.1 | 40.1 | 0.1 | **53.0** | 0.1 | 0.1 |
| | Freq | NS | 21.0 | 0.1 | 1.8 | 0.1 | 36.4 | 0.2 | 51.0 | 0.2 | - |
| | | SANS | 23.0 | 0.0 | 1.9 | 0.1 | 40.9 | 0.2 | 53.6 | 0.0 | - |
| | | T-SANS | **23.1** | 0.0 | **2.1** | 0.0 | †**41.0** | 0.1 | **53.8** | 0.0 | 0.1 |
| | Uniq | NS | 21.5 | 0.1 | 2.2 | 0.0 | 37.2 | 0.1 | 51.4 | 0.2 | - |
| | | SANS | 23.2 | 0.0 | 2.3 | 0.1 | **40.9** | 0.2 | 53.6 | 0.1 | - |
| | | T-SANS | †**23.3** | 0.1 | †**3.0** | 0.0 | 40.2 | 0.2 | †**54.4** | 0.1 | 0.5 |
| RotatE | None | NS | 47.0 | 0.1 | 42.5 | 0.2 | 48.6 | 0.2 | 55.8 | 0.3 | - |
| | | SANS | **47.2** | 0.1 | **42.6** | 0.1 | **49.1** | 0.1 | **56.7** | 0.0 | - |
| | | T-SANS | **47.3** | 0.1 | **42.6** | 0.1 | **49.1** | 0.1 | **56.7** | 0.1 | -0.01 |
| | Base | NS | 47.0 | 0.0 | 42.2 | 0.1 | 48.7 | 0.1 | 56.3 | 0.1 | - |
| | | SANS | **47.5** | 0.1 | **42.7** | 0.2 | **49.3** | 0.1 | **57.2** | 0.1 | - |
| | | T-SANS | **47.5** | 0.1 | **42.7** | 0.2 | **49.3** | 0.1 | 57.1 | 0.1 | 0.01 |
| | Freq | NS | 47.1 | 0.1 | 42.3 | 0.1 | 48.7 | 0.1 | 56.4 | 0.1 | - |
| | | SANS | **47.7** | 0.1 | †**42.9** | 0.2 | 49.6 | 0.0 | **57.4** | 0.1 | - |
| | | T-SANS | **47.7** | 0.1 | 42.8 | 0.2 | **49.7** | 0.1 | **57.4** | 0.1 | 0.1 |
| | Uniq | NS | 47.2 | 0.2 | 42.7 | 0.2 | 48.7 | 0.1 | 56.3 | 0.1 | - |
| | | SANS | 47.7 | 0.1 | †**42.9** | 0.1 | 49.6 | 0.1 | 57.2 | 0.1 | - |
| | | T-SANS | †**47.8** | 0.2 | 42.8 | 0.3 | †**49.8** | 0.1 | †**57.6** | 0.1 | 0.5 |
| HAKE | None | NS | 48.8 | 0.1 | **44.5** | 0.1 | 50.5 | 0.2 | 57.3 | 0.1 | - |
| | | SANS | **48.9** | 0.0 | **44.5** | 0.2 | **50.6** | 0.2 | 57.7 | 0.1 | - |
| | | T-SANS | **48.9** | 0.0 | 44.4 | 0.1 | 50.5 | 0.3 | **57.8** | 0.1 | 0.01 |
| | Base | NS | 49.2 | 0.0 | 44.6 | 0.1 | 51.1 | 0.1 | 57.9 | 0.2 | - |
| | | SANS | **49.5** | 0.1 | **45.0** | 0.2 | 51.2 | 0.2 | 58.2 | 0.2 | - |
| | | T-SANS | **49.5** | 0.1 | **45.0** | 0.2 | 51.2 | 0.3 | **58.4** | 0.2 | 0.1 |
| | Freq | NS | 49.3 | 0.1 | 44.8 | 0.1 | 51.3 | 0.2 | 58.0 | 0.2 | - |
| | | SANS | **49.7** | 0.1 | **45.2** | 0.2 | 51.5 | 0.1 | **58.4** | 0.2 | - |
| | | T-SANS | **49.7** | 0.0 | **45.2** | 0.2 | **51.6** | 0.3 | **58.4** | 0.2 | -0.01 |
| | Uniq | NS | 49.4 | 0.2 | 44.9 | 0.2 | 51.3 | 0.2 | 57.8 | 0.2 | - |
| | | SANS | †**49.9** | 0.0 | 45.3 | 0.1 | †**51.8** | 0.2 | †**58.6** | 0.2 | - |
| | | T-SANS | †**49.9** | 0.1 | †**45.4** | 0.1 | †**51.8** | 0.2 | 58.5 | 0.2 | 0.05 |

Table 6: Results on YAGO3-10.

| Model | Subsampling | | MRR | | H@1 | | H@3 | | H@10 | | $\gamma$ |
|---|---|---|---|---|---|---|---|---|---|---|---|
| | | | Mean | SD | Mean | SD | Mean | SD | Mean | SD | |
| RotatE | None | NS | 43.5 | 0.1 | 32.8 | 0.2 | 49.1 | 0.2 | 63.7 | 0.3 | - |
| | | SANS | **49.6** | 0.2 | 39.9 | 0.1 | 55.3 | 0.3 | **67.3** | 0.2 | - |
| | | T-SANS | **49.6** | 0.2 | **40.0** | 0.2 | **55.4** | 0.5 | 67.2 | 0.3 | -0.05 |
| | Base | NS | 44.8 | 0.1 | 34.5 | 0.3 | 50.0 | 0.2 | 64.7 | 0.2 | - |
| | | SANS | **49.6** | 0.3 | **40.1** | 0.3 | **55.2** | 0.4 | **67.4** | 0.3 | - |
| | | T-SANS | 49.5 | 0.3 | **40.1** | 0.3 | 55.0 | 0.5 | 67.3 | 0.3 | -0.05 |
| | Freq | NS | 44.8 | 0.2 | 34.5 | 0.3 | 50.0 | 0.1 | 64.7 | 0.2 | - |
| | | SANS | **49.9** | 0.2 | **40.5** | 0.3 | **55.5** | 0.5 | **67.4** | 0.3 | - |
| | | T-SANS | **49.9** | 0.2 | **40.5** | 0.3 | **55.5** | 0.5 | **67.4** | 0.2 | 0.01 |
| | Uniq | NS | 44.4 | 0.2 | 34.0 | 0.3 | 49.8 | 0.2 | 64.3 | 0.2 | - |
| | | SANS | 50.0 | 0.3 | 40.6 | 0.2 | 55.6 | 0.3 | 67.5 | 0.2 | - |
| | | T-SANS | [†]**50.1** | 0.2 | [†]**40.7** | 0.1 | [†]**55.7** | 0.3 | [†]**67.6** | 0.3 | 0.05 |
| HAKE | None | NS | 47.4 | 0.3 | 36.6 | 0.5 | 53.9 | 0.1 | 67.0 | 0.1 | - |
| | | SANS | 53.5 | 0.2 | 44.6 | 0.3 | **59.1** | 0.4 | **69.0** | 0.2 | - |
| | | T-SANS | **53.7** | 0.1 | **45.3** | 0.3 | 59.0 | 0.1 | 68.8 | 0.1 | 0.05 |
| | Base | NS | 48.8 | 0.3 | 38.4 | 0.4 | 55.0 | 0.2 | 68.1 | 0.3 | - |
| | | SANS | **54.6** | 0.2 | **46.2** | 0.3 | 59.9 | 0.2 | 69.6 | 0.2 | - |
| | | T-SANS | 54.5 | 0.2 | 45.9 | 0.3 | 59.9 | 0.2 | **69.9** | 0.1 | -0.1 |
| | Freq | NS | 49.3 | 0.2 | 39.1 | 0.3 | 55.4 | 0.1 | 68.1 | 0.2 | - |
| | | SANS | 54.6 | 0.4 | 46.0 | 0.7 | **60.2** | 0.1 | **69.6** | 0.3 | - |
| | | T-SANS | **54.8** | 0.2 | **46.4** | 0.3 | 60.1 | 0.1 | **69.6** | 0.3 | 0.05 |
| | Uniq | NS | 45.2 | 0.1 | 34.3 | 0.1 | 51.1 | 0.1 | 65.8 | 0.3 | - |
| | | SANS | [†]**55.2** | 0.3 | [†]**46.8** | 0.5 | [†]**60.5** | 0.2 | [†]**70.0** | 0.3 | - |
| | | T-SANS | 55.1 | 0.2 | [†]**46.8** | 0.3 | 60.3 | 0.1 | 69.9 | 0.2 | -0.1 |

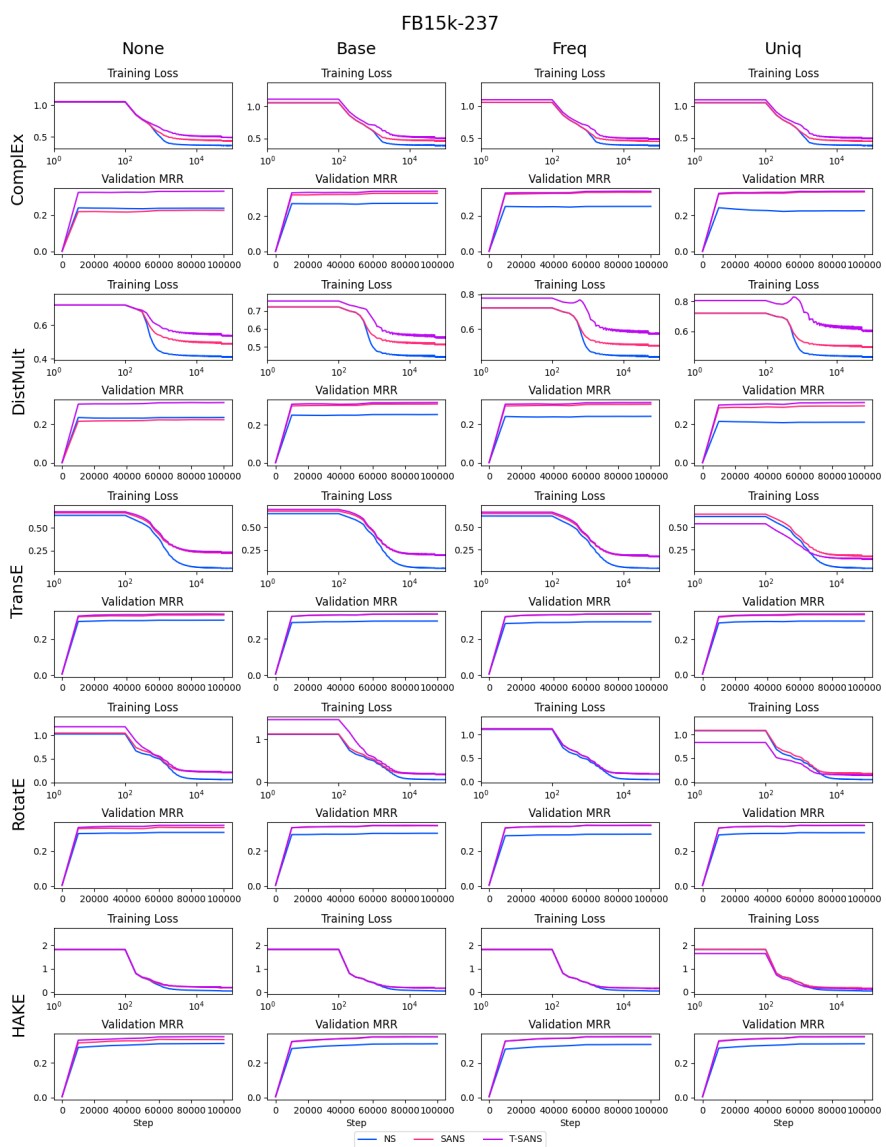

Figure 5: Training loss and validation MRR Curve on FB15k-237.

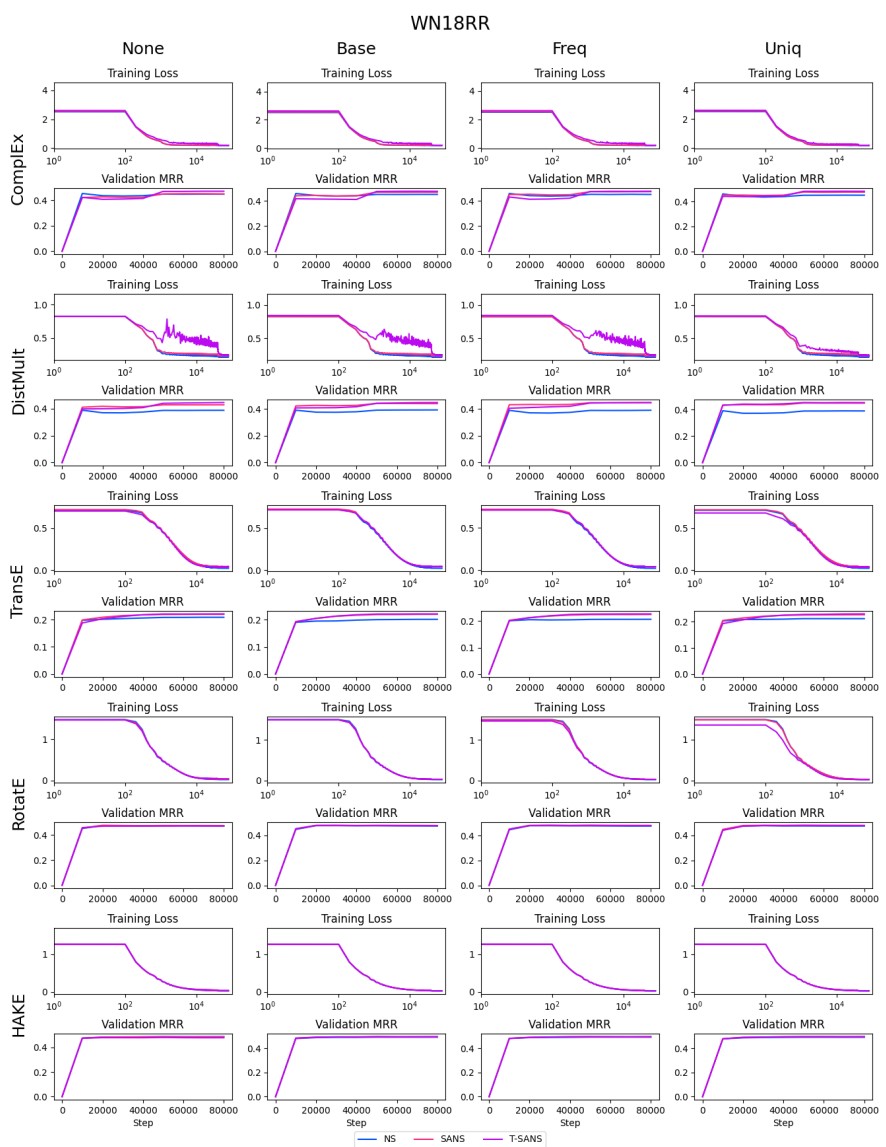

Figure 6: Training loss and validation MRR Curve on WN18RR.

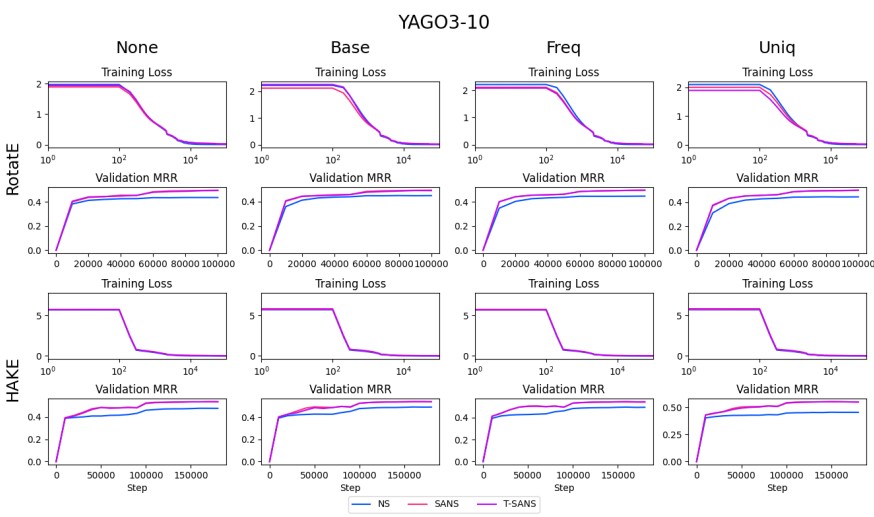

Figure 7: Training loss and validation MRR Curve on YAGO3-10.

Table 7: Statistics of the extremely sparse datasets synthesized from FB15k-237, WN18RR, and YAGO3-10.

| Dataset | Split | Tuple | Query | Entity | Relation |
|---|---|---|---|---|---|
| FB15k-237 | Total | 111,631 | 63,330 | 11,828 | 155 |
| | #Train | 95,244 | 55,923 | 11,600 | 155 |
| | #Valid | 7,571 | 6,918 | 4,933 | 90 |
| | #Test | 8,816 | 7,830 | 5,406 | 89 |
| WN18RR | Total | 14,697 | 14,675 | 12,973 | 10 |
| | #Train | 13,758 | 13,785 | 12,275 | 10 |
| | #Valid | 465 | 619 | 613 | 9 |
| | #Test | 474 | 623 | 619 | 8 |
| YAGO3-10 | Total | 366,079 | 182,274 | 95,788 | 29 |
| | #Train | 362,728 | 181,196 | 95,432 | 29 |
| | #Valid | 1,662 | 2,316 | 2,113 | 13 |
| | #Test | 1,689 | 2,359 | 2,135 | 14 |

Table 8: Performances of KGE models HAKE and RotatE on extremely sparse datasets artificially derived from FB15k-237, WN18RR, and YAGO3-10. Bold values indicate the best scores.

| Model | Loss | FB15k-237 | | | | WN18RR | | | | YAGO3-10 | | | |
|---|---|---|---|---|---|---|---|---|---|---|---|---|---|
| | | MRR | H@1 | $\gamma$ | $\beta$ | MRR | H@1 | $\gamma$ | $\beta$ | MRR | H@1 | $\gamma$ | $\beta$ |
| HAKE | NS | 38.1 | 28.4 | - | | 10.8 | 8.7 | - | | 45.9 | 36.9 | - | |
| | SANS | 35.2 | 24.5 | - | 1.0 | 10.3 | 7.8 | - | 1.0 | 47.8 | **40.0** | - | 1.0 |
| | T-SANS | **41.1** | **33.0** | -1.0 | 1.0 | **13.9** | **12.1** | -2 | 1.0 | **49.3** | **40.0** | -0.5 | 1.0 |
| RotatE | NS | 40.0 | 30.8 | - | | 14.2 | **11.8** | - | | 38.0 | 28.7 | - | |
| | SANS | 36.3 | 25.3 | - | 1.0 | 13.9 | 11.7 | - | 1.0 | 41.3 | 32.3 | - | 1.0 |
| | T-SANS | **41.5** | **33.1** | -1.0 | 1.0 | **14.4** | **11.8** | -2 | 1.0 | **45.3** | **38.1** | -0.5 | 1.0 |

