# OpenReview forum: "Unified Interpretation of Smoothing Methods for Negative Sampling Loss Functions in Knowledge Graph Embedding"
_ICLR.cc/2024/Conference — Submitted to ICLR 2024_

### Official Review · Reviewer_8iQT · 2023-10-27

**Soundness:** 2 fair
**Presentation:** 2 fair
**Contribution:** 2 fair
**Rating:** 6
**Confidence:** 2

**Summary:**

Knowledge Graphs (KGs) are vital in NLP, but creating them manually has limitations, leading to KG Completion (KGC) using KG Embedding (KGE) and Negative Sampling (NS) to handle many entities while reducing computational costs. The challenge of sparsity due to low link appearance frequencies in KGs is addressed through various smoothing methods like Self-Adversarial Negative Sampling (SANS), with this paper offering theoretical insights and introducing Triplet-based SANS (T-SANS), showing improved performance on multiple datasets.

**Strengths:**

The author illustrated clear motivation of this study and articulate the problem formulation mathematical-clearly.

The authors provided sufficient information about the background and previous studies are well summarized.

I like the plots in the introduction, better demonstrating the existing challenges.

The authors provided thorough experimentation to demonstrate the improved performance of the T-SANS, with quantitative comparison.

**Weaknesses:**

There are too many uncertainties in the contribution. The authors presented
•  We theoretically show that T-SANS with subsampling can potentially cover the conventional usages of SANS and subsampling.
• We empirically verify that T-SANS improves KGC performance on sparse KGs in terms of
MRR.
• We empirically verify that T-SANS with subsampling can cover the conventional usages of
SANS and subsampling in terms of MRR.

Without confirmative mathematical approval, it's less convincing to argue that the conventional usages of SANS and subsampling can be covered by T-SANS. And the empirical study results also weaken this argument.

Thus, I think the contribution is over claimed.

I believe the negative sampling performance can vary significantly based on the distribution of KG in real-world data. It's worthy to mention that based on what kind of distribution of KG features (e.g., connectives) the proposed method can achieve compelling performance.

**Questions:**

Is the focus of this study to improve interpretation or model performance of KGE?

What is the distribution of the entities and edges of KG in the dataset look like? Does distribution of network feature influence the performance of the proposed NS algorithm?

What are the exact performance is in the experimentation. You only present the figures. Can you include table?

---

> ### Author Response · Authors · 2023-11-14
> **Appreciating your concern: motivation and contributions of our work explained**
>
> ## Response to Reviewer @8iQT
> Dear 8iQT, we highly appreiciate your valuable comments on our work. Based on feedbacks from you, we have carefully made revisions to our paper, and we are more than committed to following your suggestions and enhance our work more.
>
> ### **Weaknesses:**
> >#### The motivation of this study is a little bit confusing. And the contribution is not clearly articulated.
> - Our motivation is to solve the problem in KGC caused by the sparsenes of KGs with our propsed theoretically appropriate smoothing method, T-SANS.
> - As stated in Section $\S1$, for doing that, we mainly contributed in:
>   - By focusing on the smoothing targets, we theoretically reveal the difference between SANS and subsampling and induce a new NS loss, Triplet-based SANS (T-SANS), that can cover the smoothing target of both SANS and subsampling.
>   - We theoretically and empirically verify how our proposed method can potentially cover the conventional methods and improve KGC performance on sparse KGs.
> - To suport the explanation of the above motivation, we added the explanation to $\S3.1$ in the updated version of our paper. Besides, the difference of T-SANS and conventional subsampling methods are discussed in $\S3.3$
>
> ### **Questions:**
> >#### Is the focus of this study to improve interpretation or model performance of KGE?
> - The main focus is to provide a systematic interpretation, by doing this, we propose the T-SANS and the unified loss function from the viewpoint of smoothing methods that formulate our findings and even help improve the performance of KGC.

---

> ### Author Response · Authors · 2023-11-22
> **OpenReview Deadline Confirmation**
>
> ### **We appreciate your insightful review and feedback comments. Now, the deadline for the discussion period only left 18.5hours. To clarify what part of your concerns remains, would it be possible for you to state them?**

---

> ### Author Response · Authors · 2023-11-22
> **Summing up our updates for this paper**
>
> Dear 8iQT,
>
> To further refine our manuscript and address the insightful feedback from all reviewers, we have made several meaningful enhancements that we hope will not only meet but exceed your expectations:
>
> 1. We've expanded Appendix B.2 to include a comprehensive analysis of the training loss and validation MRR curves for each method to provide a clearer, more nuanced understanding of our methods.
> 2. In an effort to robustly validate our approach, we've conducted additional experiments using the recent Knowledge Graph Embedding model, HousE (Li et al., 2022). These experiments yielded consistent conclusions with that in our current paper, where our T-SANS method achieving significant improvements in MRR on the FB15k-237 dataset: 24.4% over SANS and 28.6% over NS. This not only reinforces the findings in our paper but also broadens the scope of our method's applicability. We're in the process of finalizing these results, post the completion of experiments with various random seeds, and plan to incorporate them into the updated version of our paper. We anticipate that this comprehensive suite of experiments will take about one more week.
> 3. To enhance the reader's experience and understanding, we've carefully revised $\S3$, which outlines our proposed method. These revisions include not only textual improvements for better flow and comprehension but also examples to aid in grasping the concepts for researchers in other fields.
>
> Our goal with these revisions is to not only address the valuable feedback from reviewers but also to contribute a paper that is engaging, informative, and a pleasure to read. We believe these updates will substantially elevate the quality of our work and hopefully resonate positively with you.

---

### Official Review · Reviewer_4kKk · 2023-10-30

**Soundness:** 2 fair
**Presentation:** 2 fair
**Contribution:** 2 fair
**Rating:** 3
**Confidence:** 3

**Summary:**

This paper provides a unified interpretation of smoothing methods, SANS and subsampling, for negative sampling loss function in KGE. Authors emphasize the importance of smoothing both p(x,y), p(y|x), and p(x) in the loss function to deal with the data sparsity of KG. Based on the analysis of SANS and subsampling negative sampling loss function, authors propose a new negative sampling function T-SANS, which integrate both subsampling and SANS in the loss function. Experiments show that with T-SANS as the negative sampling method, the KGE models generally performs better than SANS or existing subsampling negative sampling methods, especially on the extreme unbalanced and sparse KGs.

**Strengths:**

1. The topic is interesting and worth to investigate since negative sampling methods significantly affects the KGE performance for KGC tasks.
2. The paper tried to find a uniform loss function representation for SANS and subsampling negative sampling methods, which is good.
3. T-SANS performs better than SANS and subsampling methods supports the importance of smoothing both p(x,y), p(y|x), and p(x) in the loss function.

**Weaknesses:**

1. My main concern of the paper is limited novelty contribution. T-SANS adds the subsampling based on SANS, referring to $p_{\theta}(x;\gamma)$ in Equation (12), which is the key difference between T-SANS and SANS. While as mentioned by the author in the footnote, Sun et al. (2019); Zhang et al. (2020b) use subsampling in their released implementation without referring to it in their paper. Thus, if I understood correctly,  I would like to say the actual implementation of methods of Sun et al. (2019); Zhang et al. (2020b) is very similar to T-SANS. Thus the novelty of this paper is limited.
2. The work is motivated by that conventional works use SANS and subsampling with no theoretical background, and authors believed there is room for further performance improvement. It is unclear the why the lack of the theoretical background lead to potential performance improvement.
3. Some parts of the paper is not clearly explained or inaccurate and need further improvement, such as
* the $^{-\alpha}$ in Equation (4) is unexplained
* statement in page 5 that "using Eq. (11) causes an imbalanced loss between the first and second terms since the sum of pθ (x, yi ) on ν number of negative samples is not always 1" is not accurate, since in the implementation of the model, there usually is a softmax function among over all the negative samples for a positive triple, which will make the sum of $p_{\theta} (x, y_i )$ to ν number of negative samples to 1.
* the caption of Figure 4 is the same as Figure 3.

**Questions:**

1. What is the key/significant difference between T-SANS and the actually implementation of Sun et al. (2019); Zhang et al. (2020b) methods, i.e. SANS with subsampling?
2. Should the caption of  Figure 4 to be updated?

---

> ### Author Response · Authors · 2023-11-14
> **Appreciating your attention: the novelty and contributions of our work explained**
>
> ## Response to Reviewer @4kKk
> Dear 4kKk, I appreciate you spending time reading our work and I am dedicated to enhancing my paper for readers. Would you allow me to draw your attention and ask for your additional consideration of this paper? Because your valuable insights are greatly cherished, and I believe revisiting certain aspects of our hard work could bring further clarity and great value to you too.
> Additionally, based on feedback from all reviewers, the paper has been enhanced by further meticulous revisions. I assure you that our current paper has the potential to earn your approval.
>
> ### **Weaknesses:**
> >#### My main concern of the paper is limited novelty contribution. T-SANS adds the subsampling based on SANS, referring to $p_\theta(x;y)$ in Equation (12), which is the key difference between T-SANS and SANS. While as mentioned by the author in the footnote, Sun et al. (2019); Zhang et al. (2020b) use subsampling in their released implementation without referring to it in their paper. Thus, if I understood correctly, I would like to say the actual implementation of methods of Sun et al. (2019); Zhang et al. (2020b) is very similar to T-SANS. Thus the novelty of this paper is limited.
> - T-SANS does not applied the subsampling implemented by previous work and is completely different from SANS. Please allow me to explain the differences as below:
>     - As discussed in $\S2.2$ and $\S2.3.1$, respectively, Eq. (2) denotes NS loss: \begin{equation} \ell_{\text{NS}}(\mathbf{\theta}) = -\frac{1}{|D|}\sum_{(x,y) \in D} \Bigl[\log(\sigma(s_{\mathbf{\theta}}(x,y)+\tau)) + \frac{1}{\nu}\sum_{y_{i}\sim U}^{\nu}\log(\sigma(-s_{\theta}(x,y_i)-\tau))\Bigr], \end{equation} Eq. (3) denotes conventional subsampling methods: \begin{align} &\ell_{\text{SUB}}(\mathbf{\theta}) \\ =&-\frac{1}{|D|}\!\!\sum_{(x,y) \in D} \!\!\Bigl[A(x,y;\alpha)\log(\sigma(s_{\theta}(x,y)\!+\!\tau)) \!+\!\frac{1}{\nu}\!\sum_{y_{i}\sim U}^{\nu}B(x,y;\alpha)\log(\sigma(-s_{\theta}(x,y_i)\!-\!\tau))\Bigr], \end{align} where Eq. (4) Eq. (6) Eq. (7) denotes different conventional subsampling strategies based on count in Eq. (5). From above, we can understand conventional subsampling methods mean applying term $A$ and $B$ on NS loss using counted triplets and queries frequencies. While T-SANS does not apply these couting strategies.
>     - As discussed in $\S2.3.2$ and $\S3.2$, respectively, Eq. (8) denotes SANS loss: \begin{align} &\ell_{\text{SANS}}(\mathbf{\theta}) =-\frac{1}{|D|}\sum_{(x,y) \in D} \Bigl[\log(\sigma(s_{\theta}(x,y)+\tau)) +\!\! \sum_{y_{i}\sim U}^{\nu}p_{\theta}(y_i|x;\beta)\log(\sigma(-s_{\theta}(x,y_i)-\tau))\Bigr], \end{align} Eq. (12) denotes T-SANS loss, and Eq. (13) denotes the detailed calculation for $p_{\mathbf{\theta}}(x;\gamma)$ which is innovated by us:
> \begin{align}
> &\ell_{\text{T-SANS}}(\mathbf{\theta}) \nonumber\\
> =& -\frac{1}{|D|}\sum_{(x,y) \in D} \!\!\!p_{\theta}(x;\gamma)\Bigl[\log(\sigma(s_{\theta}(x,y)+\tau)) +\!\! \sum_{y_{i}\sim U}^{\nu}p_{\theta}(y_i|x;\beta)\log(\sigma(-s_{\theta}(x,y_i)-\tau))\Bigr],\\
> &p_{\mathbf{\theta}}(x;\gamma) = \sum_{y_{i}\in D} p_{\mathbf{\theta}}(x,y_{i};\gamma),\:\:\:\:\:p_{\mathbf{\theta}}(x,y_i;\gamma) = \frac{\exp{(\gamma s_{\theta}(x,y_i))}}{\sum_{(x',y')\in D}\exp{(\gamma s_{\theta}(x',y'))}},
> \end{align} From above, we can understand that T-SANS and SANS are completely different, with SANS only taking into account the conditional probability of negative samples and T-SANS being a loss function that considers the joint probability of the pair of queries and their answers.
>     - Thus, our proposal is novel, with the key innovation in weighting through $p_{\mathbf{\theta}}(x;\gamma)$.
> - We also conducted additional experiments and illustrated the performance of T-SANS compared with conventional subsampling methods and SANS in Figure 3(a) to show their difference in essence.
> - Since the first version of our paper did not emphasize the difference between T-SANS and SANS, we added the explanation to $\S3.1$. Besides, the difference of T-SANS and conventional subsampling methods is discussed in $\S3.3$.
>
> >#### The work is motivated by that conventional works use SANS and subsampling with no theoretical background, and authors believed there is room for further performance improvement. It is unclear the why the lack of the theoretical background lead to potential performance improvement.
> - Since the previous smoothing methods are not comprehensive, the lack of theoretical background will lead to mistakes in applying a proper conventional smoothing strategy. If the smoothing strategy is not proper, the performance is not optimized. Besides, the SANS does not consider the query probability thus is weak to sparse KGs. On the contary, our proposed method are built upon a comprehensive theoretical foundation that combines both subsampling and enhanced SANS, thus can help improve the KGC performance. The experimental results showing the improvement can be seen in Figure 3,4.

---

> ### Author Response · Authors · 2023-11-14
> **Appreciating your attention: notations of our work explained**
>
> (Continuing the Response to Reviewer @4kKk)
> >#### Some parts of the paper is not clearly explained or inaccurate and need further improvement, such as: (1) the $-\alpha$ in Equation (4) is unexplained; (2) statement in page 5 that "using Eq. (11) causes an imbalanced loss between the first and second terms since the sum of $p_\theta(x,y_i)$ on $\nu$ number of negative samples is not always 1" is not accurate, since in the implementation of the model, there usually is a softmax function among over all the negative samples for a positive triple, which will make the sum of $p_\theta(x,y_i)$ to $\nu$ number of negative samples to 1. (3) the caption of Figure 4 is the same as Figure 3.
> - As explained in the first version of our paper, $\alpha$ is a temperature term to adjust the frequecy of triplets and queries. Note that we incorporate $\alpha$ into Eq. (3) to consider various loss functions even though Kamigaito & Hayashi (2022a;b) do not consider $\alpha$. Eq. (4-7) shows several assumptions for deciding Eq. (3), thus the annotation of $\alpha$ holds the same meaning as explained by us when introducing Eq. (3).
> - In Eq. (11), we mean directly calculating $p_\theta(x,y_i)$, because the score function in KGE model aims to approximate the plausibility of an example, so the most direct way to calculate plausibility in KGE models is using $p_\theta(x,y_i) = s_{\theta}(x,y_i)$, where softmax function is not applied. Instead, as we stated below Eq. (11), we choose to calculate this probability using the decomposition $p_{\mathbf{\theta}}(x,y)=p_{\mathbf{\theta}}(y|x)p_{\mathbf{\theta}}(x)$ so that the sum of $p_{\mathbf{\theta}}(y|x)$ of all negative samples is always 1 but the sum of $p_{\mathbf{\theta}}(x,y)$ is not always 1 by $p_{\mathbf{\theta}}(x)$.
> - Following your reminding, we have modified the captions for Figure 3 and 4.
> - Considering your first review is not by the accurate understanding in our paper, would it be possible to increase the score based on the updated information? Also, we hope to have the opportunity to hear more of your feedback.

---

> ### Author Response · Authors · 2023-11-22
> **OpenReview Deadline Confirmation**
>
> ### **We appreciate your insightful review and feedback comments. Now, the deadline for the discussion period only left 18.5hours. To clarify what part of your concerns remains, would it be possible for you to state them?**

---

> > ### Comment · Reviewer_4kKk · 2023-11-22
> > **Acknowledgement**
> >
> > Thanks for the clarification. I would like to keep my original score.

---

> > > ### Author Response · Authors · 2023-11-22
> > >
> > > Dear 4kKk,
> > >
> > > We appreciate your prompt response.
> > >
> > > However, this communication is not a discussion but rather a statement, as there seems to be no rationale for the decision. In your initial review, you identified six weaknesses and raised two questions regarding our paper. We have addressed each point comprehensively, including updates to the paper. As a reviewer, you are responsible for comparing your initial review with our responses and determining which parts of our response meet your criteria. Additionally, if any part of our response does not satisfy your requirements, it is necessary for you to specify why it falls short. If you cannot fulfill your duties as a reviewer due to time constraints or other commitments, kindly state it. Such an acknowledgment will aid the Area Chairs (AC) and Senior Area Chairs (SAC) in making an informed decision regarding the final review outcome.

---

### Official Review · Reviewer_s2ka · 2023-10-31

**Soundness:** 3 good
**Presentation:** 3 good
**Contribution:** 3 good
**Rating:** 5
**Confidence:** 3

**Summary:**

The paper delves into the significance of Knowledge Graphs (KGs) in Natural Language Processing (NLP) tasks. The primary focus is on Knowledge Graph Completion (KGC), which aims to automatically complete KGs by scoring their links using Knowledge Graph Embedding (KGE). The paper discusses the challenges posed by the sparsity of KGs and the role of Negative Sampling (NS) loss in addressing these challenges. The paper introduces smoothing methods like Self-Adversarial Negative Sampling (SANS) and subsampling to tackle the sparsity issue. The main contribution is a theoretical interpretation of these smoothing methods and the introduction of a new NS loss called Triplet-based SANS (T-SANS). Experimental results on various datasets demonstrate the effectiveness of T-SANS.

**Strengths:**

S1 The paper provides a comprehensive theoretical understanding of smoothing methods for NS loss in KGE.
S2 The paper presents experimental results on multiple datasets, showcasing the effectiveness of T-SANS.

**Weaknesses:**

W1 While T-SANS aims to improve upon existing methods, the computational overhead, especially in terms of memory usage and processing time, might not be thoroughly addressed.

W2 While the paper provides a comprehensive theoretical understanding of smoothing methods for NS loss in KGE, it might be too dense for a broader audience. The depth of the theoretical content might make it less accessible to practitioners or researchers from adjacent fields.

W3 How does T-SANS handle extremely sparse datasets compared to other methods? Is there a threshold of sparsity beyond which T-SANS might not be as effective?

W4 How generalizable is T-SANS to other related tasks beyond KGC? Has it been tested on tasks other than KG embedding?

**Questions:**

Q1 How does T-SANS handle extremely sparse datasets compared to other methods? Is there a threshold of sparsity beyond which T-SANS might not be as effective?

Q2 How generalizable is T-SANS to other related tasks beyond KGC? Has it been tested on tasks other than KG embedding?

---

> ### Author Response · Authors · 2023-11-14
> **Experiments on extremely sparse datasets explained and accessibility improved**
>
> ## Response to Reviewer @s2ka
> Dear s2ka, we would like to express our sincere appreciation for your invaluable feedback on our paper. We have made improvements to our paper based on your valuable input. Would it be possible to increase the score based on our additional work?
>
> ### Weaknesses:
> >#### W1 While T-SANS aims to improve upon existing methods, the computational overhead, especially in terms of memory usage and processing time, might not be thoroughly addressed.
> - Following your suggestion, we are trying to demonstrate the actual memory usage and processing time now.
>
> >#### W2 While the paper provides a comprehensive theoretical understanding of smoothing methods for NS loss in KGE, it might be too dense for a broader audience. The depth of the theoretical content might make it less accessible to practitioners or researchers from adjacent fields.
> - Please allow me to cite an example to show the difference between our T-SANS and conventional smoothing mothods: Our T-SANS can work even when the entity or relations included in the target triplet appear more than once, which is theoretically different from conventional approaches.
> - Following your suggestions, we acknowledged the importance of accessibility and added this example in $\S3.3$ to make it easier for a wider audience to comprehend and apply our methods.
>
> ### Questions:
> >#### W3 How does T-SANS handle extremely sparse datasets compared to other methods? Is there a threshold of sparsity beyond which T-SANS might not be as effective?
> - T-SANS can effectively leverage model-predicted probabilities, enabling it to handle both normally distributed and extremely sparse datasets. In contrast, baseline methods struggle to adequately consider the frequency of sparse triples, resulting in sub-optimal performance on such challenging datasets.
> - To thoroughly evaluate T-SANS, we conducted experiments on three extremely sparse datasets derived from FB15k-237, WN18RR, and YAGO3-10. As outlined in $\S6$ and illustrated in Figure 4, T-SANS consistently demonstrates improved Knowledge Graph Completion (KGC) performance in scenarios where KGs are extremely sparse. Further, by comparing with the results showcased in Figure 3, we can understand the trend that the greater the dataset sparsity, the more T-SANS outperforms conventional methods.
> - Following your viewpoint, we have added a concise comparision when analyzing in $\S6$ and moved the picture positions for better accessibility.
>
> >#### W4 How generalizable is T-SANS to other related tasks beyond KGC? Has it been tested on tasks other than KG embedding?
> - Regarding generalizability, since word embeddings and item recommendations are similar to the special case of KGs whose triplets have the same relationships, our work may inspire researchers in these fields. Test in a new task would need much more work, so we planned to put it into future work.
> - Following your viewpoint, we added this note in $\S8$ to provide more value of our work and to inspire other researchers.

---

> ### Author Response · Authors · 2023-11-17
> **Memory and time cost enhanced**
>
> Dear s2ka, for your first suggestion, we have conducted detailed analysis in the appendix of our paper.
> >#### W1 While T-SANS aims to improve upon existing methods, the computational overhead, especially in terms of memory usage and processing time, might not be thoroughly addressed.
> - Following your point of view, we have updated our paper by doing supplementary work: we analysed memory usage in $\S3.2$, and illustrated convergence curves for actual training loss and validation MRR in Appendix $\S B.2$ for each setting. By doing this, we can understand that the T-SANS has the same memory and time cost as SANS and NS.

---

> ### Author Response · Authors · 2023-11-22
> **OpenReview Deadline Confirmation**
>
> ### **We appreciate your insightful review and feedback comments. Now, the deadline for the discussion period only left 18.5hours. To clarify what part of your concerns remains, would it be possible for you to state them?**

---

> > ### Comment · Reviewer_s2ka · 2023-11-22
> > **Response to the authors**
> >
> > I appreciate the thorough responses. My opinion of the paper remains unchanged.

---

> > > ### Author Response · Authors · 2023-11-22
> > >
> > > Dear s2ka,
> > >
> > > We appreciate your prompt response.
> > >
> > > However, we must clarify that this communication is not a discussion but rather a statement, as there seems to be no rationale provided for the decision made. In your initial review, you identified four weaknesses and raised two questions regarding our paper. We have addressed each point comprehensively, including updates to the paper. As a reviewer, it is your responsibility to compare your initial review with our responses and determine which parts of our response meet your criteria. Additionally, if any part of our response does not satisfy your requirements, it is necessary for you to specify why it falls short. If you find yourself unable to fulfill your duties as a reviewer due to time constraints or other commitments, kindly state it. Such an acknowledgment will aid the Area Chairs (AC) and Senior Area Chairs (SAC) in making an informed decision regarding the final review outcome.

---

### Official Review · Reviewer_fiTR · 2023-11-01

**Soundness:** 2 fair
**Presentation:** 3 good
**Contribution:** 2 fair
**Rating:** 6
**Confidence:** 4

**Summary:**

This paper investigates how different smoothing methods affect the negative sampling losses for knowledge graph embedding. It introduces a new triplet-based self-adversarial negative sampling method that can adjust the frequencies of triplets, queries, and answers in the training data. It evaluates the proposed method on three benchmark datasets with five base models and demonstrates its effectiveness.

**Strengths:**

1. Negative sampling is a crucial technique for learning KG embeddings. This paper offers a valuable insight into the smoothing methods for learning loss in KGE. I think it is an interesting and relevant work for the KGE community.

2. Based on the comparison and analysis of existing smoothing methods, the paper proposes triplet-based SANS, which can outperform other baselines on three datasets.

**Weaknesses:**

1. In my view, the proposed method is incremental work based on previous studies. It is an extension of SANS.

2. Another weakness is that the selected KGE models in the experiments are old. Some popular or recent models, such as TuckER [1] and HousE [2], are not included, which, in my view, may weaken the soundness the work.

[1] Ivana Balazevic, Carl Allen, Timothy M. Hospedales: TuckER: Tensor Factorization for Knowledge Graph Completion. EMNLP/IJCNLP (1) 2019: 5184-5193

[2] Rui Li, Jianan Zhao, Chaozhuo Li, Di He, Yiqi Wang, Yuming Liu, Hao Sun, Senzhang Wang, Weiwei Deng, Yanming Shen, Xing Xie, Qi Zhang: HousE: Knowledge Graph Embedding with Householder Parameterization. ICML 2022: 13209-13224

**Questions:**

1. Why are some results on YGAO3-10 missing? I think it would be better to produce the results using open-source implementations.

2. Is it possible to provide any analysis or experimental results to assess the effect of negative sampling on the convergence rate?

---

> ### Author Response · Authors · 2023-11-14
> **Novelty of our work**
>
> ## Response to Reviewer @fiTR
> Dear fiTR, we greatly appreciate your reviewing our paper, and we are committed to improve following your insightful suggestions.
>
> ### **Weaknesses:**
> >#### 1. In my view, the proposed method is incremental work based on previous studies. It is an extension of SANS.
> - Although the name T-SANS gives the impression that it is just an extension of SANS, this name was given out of respect for SANS. In fact, they are completely different, with SANS only taking into account the conditional probability of negative samples and T-SANS being a loss function that considers the joint probability of the pair of queries and their answers. This difference is large from the viewpoint of loss functions [1].
>     - [1]: Zhuang Ma, Michael Collins. "Noise Contrastive Estimation and Negative Sampling for Conditional Models: Consistency and Statistical Efficiency", https://aclanthology.org/D18-1405/
> - We conducted experiments and illustrated the performance differences of T-SANS compared with SANS in Figure 3, which verified their difference in the actual data.
> - However, Reviewer @4kKk also misunderstood this point, so we added a concise explanation emphasizing the difference in $\S3.1$.
>
> >#### 2. Another weakness is that the selected KGE models in the experiments are old. Some popular or recent models, such as TuckER [1] and HousE [2], are not included, which, in my view, may weaken the soundness the work.
> - Following your suggestion, we are trying to implement TuckER and HouseE now. Would it be possible to increase the score based on the results of the additional experiments?
>
> ### **Questions:**
> >#### 1. Why are some results on YGAO3-10 missing? I think it would be better to produce the results using open-source implementations.
> - Since YAGO3-10 is larger than FB15k-237 and WN18RR, it requires a lot of computational resources. To deal with this problem, we made the decision not to train the YAGO3-10 dataset on all models but only on two best-performing models whose efficient hyperparameters on YAGO3-10 are provided by previous work.
>
> >#### 2. Is it possible to provide any analysis or experimental results to assess the effect of negative sampling on the convergence rate?
> - Following your valuable suggestion, we believe that this analysis will further provide valuable insights and enhancing the credibility of our work, thus we are wholeheartedly committed to conducting this analysis. Would it be possible to increase the score based on the results of the additional analysis?

---

> ### Author Response · Authors · 2023-11-17
> **Convergence analysis conducted in Appendix B.2**
>
> Dear fiTR, for your second qustion, we have conducted additional analysis.
>
> >#### Is it possible to provide any analysis or experimental results to assess the effect of negative sampling on the convergence rate?
> - By following your valuable suggestion, as is refered to in $\S5.2$ of the updated revision of our paper, we conducted convergence analysis in Appendix $\S B.2$ using Figures 5,6, and 7 of training loss curves and validation MRR curves for each smoothing method for each experimental setting.
> - From Figure 5,6,and 7, we can understand that the convergence of our T-SANS loss is as well as SANS and NS loss on datasets FB15k-237, WN18RR, and YAGO3-10 for each KGE model.
> - We believe that this analysis will further underscore our dedication to providing valuable insights and enhancing the credibility of our work. Thank you again!

---

> ### Author Response · Authors · 2023-11-20
> **Our methods verified on HousE (Li et al., 2022)**
>
> Dear fiTR, for your second suggestion, we have verified our method by conducting addtional expeirments for HousE.
> >#### Another weakness is that the selected KGE models in the experiments are old. Some popular or recent models, such as TuckER [1] and HousE [2], are not included, which, in my view, may weaken the soundness the work.
> - We would like to share with you that we have verified our method for the most recent KGE model HousE (Li et al., 2022), and that the experimental results show that our T-SANS gains 24.4% and 28.6% improvement in MRR on dataset FB15k-237 compared with SANS and NS, respectively. Thus, the conclusion is consistent with that in our paper.
> - We are also conducting implementation for TuckER (Balazevic et al., 2019).
> - However, because of the computing resource limitation, we will need one more week to finish all the experiments. If our paper is accepted, I am sure that we will show all the experimental results expected. Would it be possible to increase the score based on the results of the additional experiments?

---

> > ### Author Response · Authors · 2023-11-23
> > **Adaptation of our method on TuckER (Balazevic et al., 2019) might be less relevant or meaningful**
> >
> > Dear fiTR,
> >
> > Due to the inherent design of the TuckER (Balazevic et al., 2019), which involves calculating probabilities of every node for each triple, it essentially conflicts with the concept of negative sampling. Implementing negative sampling and subsampling in TuckER would disrupt its original design, making such adaptations less relevant or meaningful. So we may not implement TuckER for our method.

---

> ### Author Response · Authors · 2023-11-22
> **OpenReview Deadline Confirmation**
>
> ### **We appreciate your insightful review and feedback comments. Now, the deadline for the discussion period only left 18.5hours. To clarify what part of your concerns remains, would it be possible for you to state them?**

---

### Meta-Review · Area_Chair_dYqd · 2023-12-05

**Metareview:**

The paper presents an analysis of smoothing methods in Negative Sampling (NS) loss functions for Knowledge Graph Completion (KGC) and introduces a new NS loss, Triplet-based SANS (T-SANS). It theoretically interprets smoothing methods and tests T-SANS on various datasets, demonstrating its effectiveness. However, reviewers raised significant concerns: the incremental nature of T-SANS over existing methods, exclusion of recent KGE models in experiments, potential computational overhead, and limited accessibility of the content due to its dense theoretical nature. Additionally, questions about T-SANS's handling of extremely sparse datasets, its generalizability beyond KGC, and the overall novelty and contribution of the work were noted. These weaknesses suggest that further refinement and exploration are needed to fully establish the merits and applications of T-SANS.

**Justification For Why Not Higher Score:**

The paper is incremental.

**Justification For Why Not Lower Score:**

NA

---

### Decision · Program_Chairs · 2024-01-16

Reject